# LOOKING BACKWARD: STREAMING VIDEO-TO-VIDEO TRANSLATION WITH FEATURE BANKS

**Feng Liang** [1]   **Akio Kodaira** [2]   **Chenfeng Xu** [2]
**Masayoshi Tomizuka** [2]   **Kurt Keutzer** [2]   **Diana Marculescu** [1]
[1] UT Austin                    [2] UC Berkeley
{jeffliang, dianam}@utexas.edu, {akio.kodaira, xuchenfeng}@berkeley.edu
https://jeff-liangf.github.io/projects/streamv2v

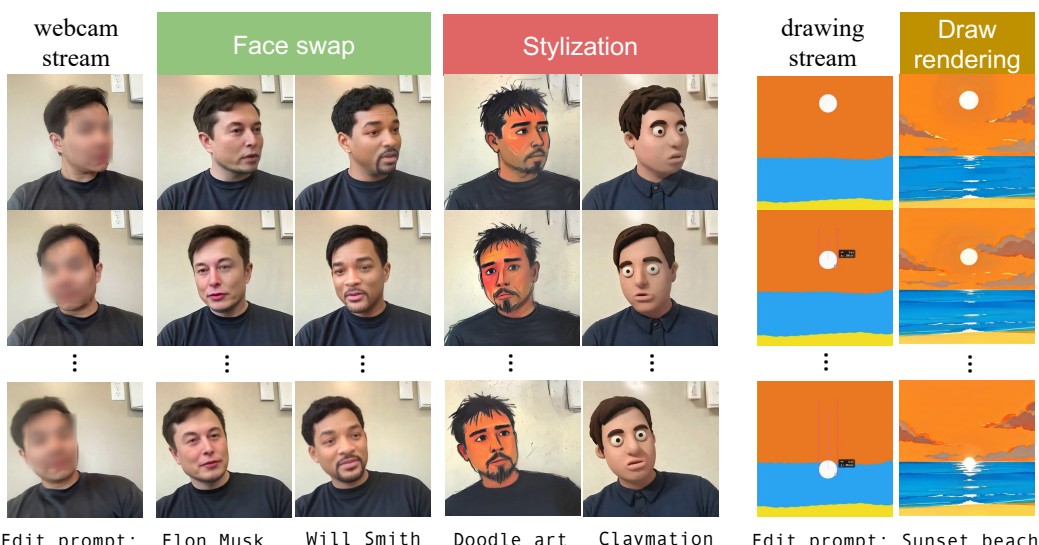

Figure 1: We present StreamV2V to support real-time video-to-video translation for streaming input. For webcam input, our StreamV2V supports face swap (*e.g.*, to Elon Musk) and video stylization (*e.g.*, to doodle art). Additionally, StreamV2V provides drawing rendering capabilities, enabling iterative creation. We encourage readers to check our video results in the supplementary materials.

## ABSTRACT

This paper introduces StreamV2V, a diffusion model that achieves real-time streaming video-to-video (V2V) translation with user prompts. Unlike prior V2V methods using batches to process a limited number of frames, we opt to process frames in streaming fashion, to support an unlimited number of frames. At the heart of StreamV2V lies a *backward-looking* approach that relates the present to the past. This is realized by maintaining a feature bank that archives information from past frames. For incoming frames, StreamV2V extends self-attention to include banked keys and values, and directly fuses similar past features into the output. The feature bank is continually updated by merging stored and new features, making it compact yet informative. StreamV2V stands out for its adaptability and efficiency, seamlessly integrating with image diffusion models without fine-tuning. StreamV2V can run 20 FPS on one A100 GPU, being 15×, 46×, 108×, and 158× faster than FlowVid, CoDeF, Rerender, and TokenFlow, respectively. Quantitative metrics and user studies confirm StreamV2V's exceptional ability to maintain temporal consistency. Demo, code, and models are available on the project page.

## 1 INTRODUCTION

Text-driven video-to-video (V2V) translation, which aims to alter the input video following given prompts, has wide applications, such as creating short videos, and more broadly in the film industry. Most diffusion model based methods (Wu et al., 2023b; Yang et al., 2023; Ouyang et al., 2023; Wang et al., 2023a; Khachatryan et al., 2023; Qi et al., 2023; Zhang et al., 2023; Wang et al., 2023b; Chen

et al., 2023; Zhao et al., 2023; Geyer et al., 2023; Liang et al., 2023; Wu et al., 2023a; Singer et al., 2024) use batches to process recorded videos, as shown in Figure 2(a). However, batch processing necessitates loading all frames into GPU memory, thereby limiting the video length they can handle, typically up to 4 seconds. Furthermore, these methods do not accommodate real-time translation and typically require several minutes to process a single 4-second clip.

This paper targets streaming V2V applications, such as webcam video translation and AI-assisted drawing, where users want to modify the streaming video iteratively. This necessitates the model to handle input videos of varying lengths and perform real-time translation. To tackle this challenge, we introduce StreamV2V, an approach that processes frames in streaming fashion, as shown in Figure 2(b). Leveraging advancements in one-/few-step diffusion models (Song et al., 2023; Sauer et al., 2023; Luo et al., 2023b), StreamDiffusion (Kodaira et al., 2023) has designed a pipeline for real-time interactive image generation. However, directly applying StreamDiffusion for V2V tasks leads to noticeable pixel flickering across frames. This is because StreamDiffusion treats each frame independently, disregarding the continuity of videos. In contrast, humans implicitly memorize visual elements across frames and see the current frame as it links with past observations. To generate consistent videos, it is critical to integrate a mechanism that can effectively bridge the current frame to its predecessors.

(a) Batch processing  (b) Stream processing

(c) Memory consumption comparsion

Figure 2: (a) Existing V2V methods process frames in batches, restricting them to a limited number of frames. (b) Our StreamV2V framework processes frames in streaming fashion, can operate on streaming videos in real-time. (c) Batch processing requires $O(N)$ memory for the video length $N$, whereas our StreamV2V only needs $O(1)$ memory regardless of video length.

Recent studies have shown that diffusion features (Tang et al., 2024; Luo et al., 2023a) captured during U-Net's forward process contain rich correspondences between images. Inspired by this, our StreamV2V maintains a feature bank, which stores the intermediate features of past frames. For incoming frames, we extend self-attention by incorporating the corresponding stored keys and values. This can be interpreted as a weighted sum of similar regions across frames via attention, effectively aligning the current frame with previous frames. Additionally, to ensure the consistency of fine-grained details, we directly fuse the block's output with similar features from past frames.

The challenge then becomes: *How can we implement this feature bank?* A straightforward approach might store a constant number of frames, such as employing a sliding window technique. However, this method is sub-optimal, as it inadvertently discards valuable data when a frame is omitted, and generates redundancy when the stored frames are similar. To address this, we propose to continuously update the bank by merging redundant features within incoming and stored features. This allows us to preserve the most representative features while keeping a consistent bank size over time. Through our experiment, we find the feature bank can be condensed to the size needed to store just one frame.

StreamV2V requires no training or fine-tuning, making it compatible as an add-on with any image diffusion models. It excels in efficiency, capable of processing high-resolution video ($512 \times 512$) in real-time at 20 frames per second (FPS) on a single A100 GPU. This speed surpasses current V2V methods—FlowVid (Liang et al., 2023), CoDeF (Ouyang et al., 2023), Rerender (Yang et al., 2023), and TokenFlow (Geyer et al., 2023)—by factors of $15\times$, $46\times$, $108\times$, and $158\times$, respectively. We evaluate our method with quantitative metrics, such as CLIP score (Radford et al., 2021) and warp error (Lai et al., 2018), and a user study. Our findings indicate that users significantly favor our StreamV2V over StreamDiffusion (Kodaira et al., 2023) (with over 70% win rates) and CoDeF (Ouyang et al., 2023) (with over 80% win rates). While our method may not yet match the performance of state-of-the-art (SOTA) methods like TokenFlow and FlowVid, its rapid real-time execution opens up new venues for streaming V2V applications.

Our contributions are three-fold: (1) To the best of our knowledge, our approach is the first approach to tackle real-time video-to-video translation for streaming videos. (2) Our method, StreamV2V,

employs a simple yet effective looking-backward principle by maintaining a feature bank to improve consistency. (3) We develop a dynamic feature bank updating strategy that merges redundant features, ensuring the feature bank remains both compact and descriptive.

## 2 RELATED WORK

### 2.1 VIDEO-TO-VIDEO TRANSLATION

Significant progress has been made in the domain of text-guided image-to-image (I2I) translation (Brooks et al., 2023; Hertz et al., 2022; Tumanyan et al., 2023; Mou et al., 2023), greatly supported by large pre-trained text-to-image diffusion models (Ramesh et al., 2022; Rombach et al., 2022; Saharia et al., 2022). Similarly, video-to-video (V2V) translation, which aims to generate consistent videos, has attracted substantial interest as well. To generate coherent multiple frames, most existing works (Esser et al., 2023; Wu et al., 2023b; Wang et al., 2023a; Guo et al., 2023; Chen et al., 2023; Khachatryan et al., 2023; Ceylan et al., 2023; Qi et al., 2023; Geyer et al., 2023; Wu et al., 2023a; Liang et al., 2023; Ku et al., 2024; Singer et al., 2024) process batches of frames simultaneously with cross-frame attention mechanisms. However, as the memory usage increases with an increased number of frames, these methods are typically constrained to about 4 seconds length. Additionally, they tend to rely on expensive DDIM inversion (Song et al., 2020; Qi et al., 2023; Geyer et al., 2023) or optical flow computation (Yang et al., 2023; Liang et al., 2023), leading to long processing time. In contrast, our StreamV2V handles videos in real-time and in streaming fashion, allowing for processing videos of any length. Compared to concurrent work Live2Diff (Xing et al., 2024) that requires additional video fine-tuning to train uni-directional temporal attention, our method is training-free and can serve as an add-on for any image diffusion model.

### 2.2 ACCELERATING DIFFUSION MODELS

While achieving great generation quality, diffusion models are commonly limited by their slow speed due to the need for multiple denoising steps. Recent advancements have introduced reusing cached features in denoising steps (Ma et al., 2023; Wimbauer et al., 2023) and one-/few-step diffusion models through distillation (Song et al., 2023; Meng et al., 2023; Luo et al., 2023b; Sauer et al., 2023; Yin et al., 2023; Lin & Yang, 2024). StreamDiffusion (Kodaira et al., 2023) proposes a pipeline to leverage these developments for real-time image generation. However, its application to video without adjustments brings unsatisfactory results. Leveraging StreamDiffusion's groundwork, we enhance frame consistency by implementing a backward-looking feature bank. Our approach introduces a dynamic merging technique for the feature bank, ensuring it remains compact and incurs minimal additional computational cost in comparison to StreamDiffusion.

### 2.3 FEATURE BANKS

Long-term feature banks (Wu et al., 2019; Pan et al., 2021) have been used in video understanding as supportive context features to help reason the entire video. Adapting this concept, we employ feature banks to enhance video generation consistency. To ensure our feature bank remains both informative and compact, we introduce a dynamic feature merging strategy. While token merging (Bolya et al., 2022; Bolya & Hoffman, 2023) is a common method for merging similar features within single images, our approach extends this technique across video frames, differentiating it from traditional within-image operations.

## 3 BACKGROUND: STREAMDIFFUSION

StreamDiffusion (Kodaira et al., 2023) leverages the Latent Consistency Model (LCM) (Luo et al., 2023b) to implement a stream batch strategy, enabling the real-time generation of images. Instead of waiting for one image to be entirely denoised (usually 2-4 steps for LCM), stream batch reformulates sequential denoising steps into batched processes. This allows simultaneous processing of $S$ images at varying denoising steps, where $S$ is the number of denoising steps. For instance, at a given timestep $t$ (assuming $t > S$), StreamDiffusion first encodes the image $I_t$ with a variational autoencoder (VAE) and adds a certain level of noise. We denote encoded latent as $z_t^S$, where the subscript $_t$ denotes the frame timestep and the superscript $^S$ denotes the denoising step. StreamDiffusion processes latent denoising batch $\{z_t^S, z_{t-1}^{S-1}, ..., z_{t-S+1}^1\}$. Upon advancing to timestep $t+1$, the model outputs the

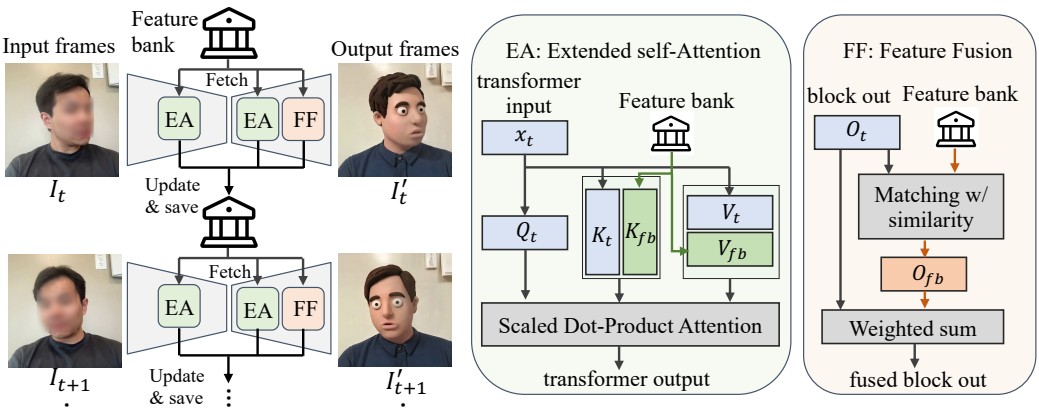

Figure 3: **Overview of StreamV2V.** Left: StreamV2V connects the current frame to the past by maintaining a feature bank that stores the intermediate transformer features. For new incoming frames, StreamV2V fetches the stored features and uses them by Extended self-Attention (EA) and direct Feature Fusion (FF). Middle: EA concatenates the stored keys $K_{fb}$ and values $V_{fb}$ directly to that of the current frame in the self-attention computation (Section 4.1). Right: Operating on the output of transformer blocks, FF first retrieves the similar features in the bank via a cosine similarity matrix, and then conducts a weighted sum to fuse them (Section 4.2). The update method of the feature bank is elaborated in Section 4.3.

final latent $z_{t-S+1}^0$, which is then decoded into the output image $I'_{(t-S+1)}$. The remaining latent would be denoised one step further, and the latent $z_{t+1}^S$ from the new image $I_{t+1}$ would be added to the batch. We include the diagram of the stream batch (Figure 12) in Appendix A.1. StreamDiffusion kick-starts the process by initializing the batch with $S$ identical starting images, enabling a warm start. To further accelerate the inference, SteamDiffusion also utilizes the Tiny autoencoder (madebyollin, 2023) and TensorRT (NVIDIA, 2024) acceleration.

## 4  STREAMV2V

While being real-time, directly applying StreamDiffusion to video-to-video generation tasks brings unsatisfactory flickering results because each frame is generated independently. Built upon StreamD-iffusion, we introduce a *backward-looking* mechanism so that the generation of the current frame can reason about the past to bring a consistent output. This is realized by maintaining a feature bank storing the information of past frames. As shown on the left of Figure 3, StreamV2V fetches the stored features in the bank to process the frame $I_t$. We introduce two training-free techniques to leverage the stored features, namely Extended self-Attention (EA) (Section 4.1) and direct Feature Fusion (FF) (Section 4.2) Lastly, we discuss how to update the compact but informative feature bank by dynamically merging the stored and newly incoming features in Section 4.3.

### 4.1  EXTENDED SELF-ATTENTION

We mainly use Stable Diffusion (Rombach et al., 2022), which is built upon the U-Net architecture. The model contains multiple encoder and decoder blocks. Each block comprises a residual convolu-tional unit and a transformer module, the latter including a self-attention layer, a cross-attention layer, and a feed-forward network. While it's been a common practice to inflate the self-attention layer to cross-frame attention in image diffusion (Hertz et al., 2023; Tewel et al., 2024; Zhou et al., 2024) or video diffusion methods (Ho et al., 2022; Wu et al., 2023b; Khachatryan et al., 2023; Ceylan et al., 2023; Qi et al., 2023; Liang et al., 2023), these approaches require processing all the frames at the same time in a batch. We extend the self-attention to accommodate the feature bank, which stores past frame information. As shown in the middle of Figure 3, for a given video frame $I_t$, we obtain the projected queries $Q_t \in \mathbb{R}^{n \times d}$, keys $K_t \in \mathbb{R}^{n \times d}$, and values $V_t \in \mathbb{R}^{n \times d}$ from the intermediate transformer input $x_t$. Here, $n$ and $d$ denote the number and dimension of feature tokens, respectively.

Denoting $K_{fb} \in \mathbb{R}^{m \times d}$, and $V_{fb} \in \mathbb{R}^{m \times d}$ as stored keys and values from previous frames, where $m$ indicates the size of the bank, we can write the extended self-attention:

$$\text{ExAttn} = \text{softmax}\left(\frac{Q_t \cdot [K_t, K_{fb}]^{Tr}}{\sqrt{d}}\right)[V_t, V_{fb}] \tag{1}$$

$[\cdot]$ and $^{Tr}$ denotes concatenation and transpose operation. Essentially, this extended self-attention functions as a weighted sum of similar areas across frames, thereby aligning the current frame with its past for improved consistency. (More illustrations can be found in Appendix A.2) We extend all the self-attention layers with all denoising times. Further details on updating the feature bank are included in Section 4.3.

## 4.2 FEATURE FUSION

While extended self-attention offers significant improvements in consistency, it operates *implicitly* through attention. We further introduce an *explicit* strategy for enhancing fine-grained consistency by directly fusing features based on correspondence. This is motivated by recent findings that diffusion features (Tang et al., 2024; Luo et al., 2023a) during the U-Net forward process contain rich correspondences between images. As shown on the right of Figure 3, for a given video frame $I_t$, we obtain the output features for the intermediate blocks $O_t \in \mathbb{R}^{n \times d}$, and we also maintain the output features of past frames $O_{fb} \in \mathbb{R}^{m \times d}$. For the token at position $p$ in $O_t$, we seek the closed token at position $q$ in $O_{fb}$, utilizing cosine similarity as described by Tang et al. (2024). We denote $O_t(p)$ as selecting the token $p$ from $O_t$ and $O'_t$ as the fused output feature:

$$O'_t(p) = (1 - \alpha)\,O_t(p) + \alpha\,O_{fb}(q),\ \text{where } q = \arg\max\left(O_t(p) \cdot O_{fb}^{Tr}\right) \tag{2}$$

where $\alpha$ is a hyperparameter to identify the strength of fusion, which is usually set to 0.75. Intuitively, this direct feature fusion aims to enhance consistency by merging similar regions from past frames to the current frame. In some cases, the current frame introduces novel regions which are absent in past frames. To prevent misalignment, we generate a mask based on a predefined similarity threshold. Specifically, we generate the mask by calculating the cosine similarities between each feature in the current output and all features stored in the bank. For each feature, we identify the most similar stored feature and calculate the similarity score. If the score is lower than a predefined threshold (set at 0.9), it indicates that we cannot find a suitable match in the feature bank. In such cases, we mask out this position, meaning this feature remains unchanged and is not fused with others from the bank. Our analysis further indicates that the location of feature fusion across various network blocks significantly impacts overall performance. Specifically, we observed that limiting feature fusion to the low-resolution decoder blocks results in optimal performance enhancements. For a comprehensive comparison and further insights, refer to Appendix A.3.

## 4.3 UPDATING THE FEATURE BANK WITH DYNAMIC MERGING

We denote all transformer intermediate features of frame $I_t$ as $T_t = \{K_t, V_t, O_t\}$, where $K_t$ and $V_t$ are projected keys and values of self-attention input, $O_t$ is the output of transformer block. An initial approach for creating a feature bank is to store features from a constant number of frames. As depicted in Figure 4 (a), after deciding the maximum number of frames (which is set to 2), the bank operates like a queue: as new frames arrive, the oldest frames are popped out to make space for the newcomers. Yet, this method encounters two primary limitations: (1) the continuous removal of the oldest frames restricts the feature span to a brief temporal window, and (2) the redundant features in the bank incur extra storage and processing costs.

To create a compact and informative bank, we propose to dynamically merge (DyMe) the stored features and newly

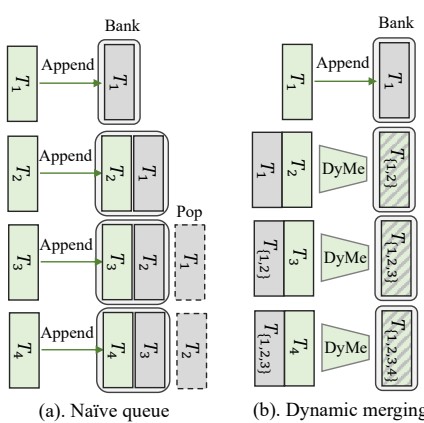

(a). Naïve queue  (b). Dynamic merging

Figure 4: **Naive queue *vs.* our dynamic merging (DyMe).** DyMe has a more compact and informative feature bank.

coming features. As shown in Figure 4 (b), at timestep 2, we merge $T_1$ and $T_2$ into $T_{\{1,2\}}$, which has the same size as $T_1$. By applying DyMe, we can the get condensed bank $T_{\{1,2,3,4\}}$ which contains the information for all 4 frames, yet occupies only half the space of a naive queue. We follow the efficient bipartite matching technique (Bolya et al., 2022; Bolya & Hoffman, 2023) to do the merging. In more detail: (1) we first concatenate features of the current frame (green boxes) and features stored in the bank (gray boxes); (2) we then randomly partition the features into two sets source (src) and destination (dst) of equal size; (3) For each feature vector $f_{\text{src}}$ in the src set, we identify the most similar feature vector $f_{\text{dst}}$ in the dst set using the cosine similarity metric defined as follows: $\text{sim}(f_{\text{src}}, f_{\text{dst}}) = \frac{f_{\text{src}} \cdot f_{\text{dst}}}{|f_{\text{src}}||f_{\text{dst}}|}$. In this expression, $\cdot$ represents the dot product, and $|\cdot|$ denotes the norm of a vector. (4) Once we find the most similar features $f_{\text{dst}}$ of $f_{\text{src}}$, we proceed to merge the features from src into dst. This is achieved by averaging the values of each matched pair: $f_{\text{dst}}^{\text{new}} = \frac{f_{\text{src}} + f_{\text{dst}}}{2}$. This step ensures that the features in dst are updated to reflect a blend of both the original and the matched features from src. Our experiments show that our dynamically merged feature bank brings better performance-speed trade-off than the naive queue-based bank. Empirically, we find the feature bank can be condensed to the size of storing just one frame. For more details, please refer to Section 5.4.2. We also visualize the effect of DyMe in Appendix A.5.1.

# 5 EXPERIMENTS

## 5.1 IMPLEMENTATION DETAILS

We built our method on StreamDiffusion (Kodaira et al., 2023) with Latent Consistency Model (Luo et al., 2023b). By default, we use a 4-step LCM without the classifier-free guidance (Ho & Salimans, 2022). We update the feature bank every 4 frames. The underlying image-to-image method is SDEdit (Meng et al., 2021), with an initial noise strength of 0.4. Unlike some methods (Yang et al., 2023; Liang et al., 2023) which use frame interpolation to generate high FPS video, our StreamV2V generates *all* frames in the same pipeline.

Following TokenFlow (Geyer et al., 2023) and FlowVid (Liang et al., 2023), we build our user study by selecting 19 object-centric videos from the DAVIS trainval 2017 dataset (Pont-Tuset et al., 2017), covering diverse subjects such as humans and animals. We reuse 67 prompts from (Liang et al., 2023), ranging from stylization to object swaps, for these videos. We conduct a thorough comparison with state-of-the-art V2V methods such as Rerender (Yang et al., 2023), CoDeF (Ouyang et al., 2023), TokenFlow (Geyer et al., 2023), and FlowVid (Liang et al., 2023), utilizing their official codes under default settings. We report both qualitative comparison 5.2, and quantitative metrics 5.3, such as CLIP score (Radford et al., 2021), warp error (Lai et al., 2018), and user preference, to verify the effectiveness of our method.

## 5.2 QUALITATIVE RESULTS

In Figure 5, we qualitatively compare our StreamV2V with several representative V2V methods, starting with our per-frame baseline StreamDiffusion (Kodaira et al., 2023), which treats each frame independently. StreamDiffusion often results in noticeable flickering, such as the background flowers and the dancer's legs. CoDeF (Ouyang et al., 2023) tends to produce outputs with significant blurriness, especially when there is a big motion within the input video, which fails in the construction of the canonical image. Rerender (Yang et al., 2023) fails to keep tracking the clothing color in the dance, which fluctuates between brown and blue. TokenFlow (Geyer et al., 2023) occasionally struggles to follow the prompt, such as transforming the video into pixel art. FlowVid (Liang et al., 2023) maintains good prompt alignment but similarly struggles with color consistency in clothing and suffers from blurriness in the final frame. In contrast, our method stands out in terms of editing capabilities and overall video quality, demonstrating superior performance over these methods. Moreover, our StreamV2V can handle arbitrary lengths of videos as showcased in the Appendix A.6.

## 5.3 QUANTITATIVE RESULTS

### 5.3.1 EVALUATION METRICS

**CLIP score** Following previous research (Geyer et al., 2023; Liang et al., 2023), we first utilize CLIP (Radford et al., 2021) to evaluate the consistency of generated videos. Specifically, we first get

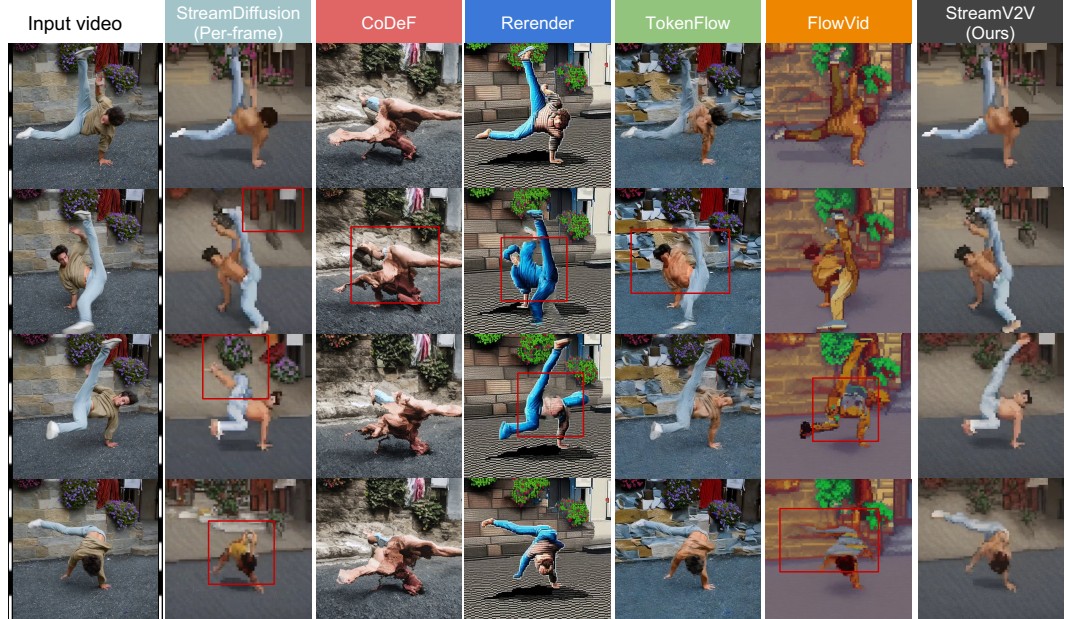

Figure 5: **Qualitative comparison with representative V2V models.** Prompt is `'A pixel art of a man doing a handstand on the street'`. Our method stands out in terms of prompt alignment and overall frame consistency. We highly encourage readers to refer to video comparisons in our supplementary videos.

Table 1: **Quantitative metrics comparison.** We report the CLIP score and warp error to indicate the consistency of generated videos. We bold the **best** result and underline the second best.

|  | StreamDiffusion | CoDeF | Rerender | TokenFlow | FlowVid | StreamV2V (ours) |
|---|---|---|---|---|---|---|
| CLIP score ↑ | 95.24 | 96.33 | 96.20 | **97.04** | 96.68 | 96.58 |
| Warp error ↓ | 117.01 | 116.17 | 107.00 | 114.25 | 111.09 | **102.99** |

the CLIP image embeddings for all the video frames, then we measure the mean cosine similarity across all sequential frame pairs. Our evaluation, detailed in Table 1, includes an analysis of 67 video-prompt pairs from the DAVIS dataset. TokenFlow shows superior performance in maintaining temporal consistency which aligns with the findings from our user study. StreamDiffusion, which is a per-frame image model, achieves the worst score. Our StreamV2V ranks in third place regarding CLIP score. Taking the consideration that our StreamV2V is a real-time stream processing method for unlimited length videos which is significantly faster than these batch processing, limited length video V2V methods (Section 5.3.2), our method is shown to have a good performance-speed trade-off.

**Warp error** We also propose to use warp error (Lai et al., 2018) as a measure of temporal consistency following previous research (Ceylan et al., 2023; Geyer et al., 2023). We first compute the optical flow between consecutive frames in input video using a pre-trained RAFT flow estimator (Teed & Deng, 2020). Then we warp the frame in the transferred video to the next using the estimated flow. This allows for the evaluation of pixel discrepancies between the warped frame and the target counterpart. We calculate the average mean squared pixel error over the un-occluded regions as warp error. As shown in Table 1, our StreamV2V achieves the best warp error among all methods.

### 5.3.2 PIPELINE RUNTIME

We benchmark the runtime with a $512 \times 512$ resolution video containing 120 frames (4 seconds video with FPS of 30) in Figure 6. For StreamV2V, we conducted ten runs, discarded the top and bottom two results, and averaged the remaining six. The runtime for FlowVid, CoDeF, Rerender, and TokenFlow is taken from the FlowVid paper (Liang et al., 2023). FlowVid (Liang et al., 2023) and Rerender (Yang et al., 2023) first generate key frames, then use frame interpolation methods (Jamriska, 2018; Huang et al., 2022) to generate non-key frames. We choose the keyframe interval of 4 for

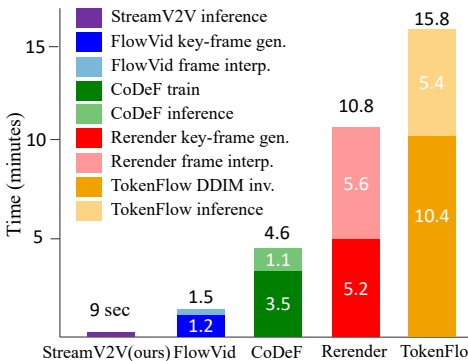

Figure 6: **Runtime breakdown** on one A100 GPU of generating a 4-second 512x512 resolution video with 30 FPS.

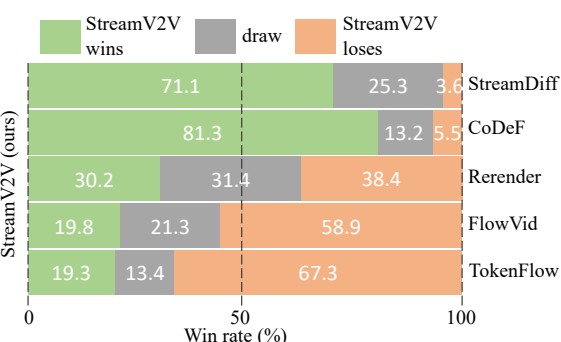

Figure 7: **User study comparison.** The win rate indicates the frequency our StreamV2V is preferred compared with certain counterpart.

these two methods, following (Liang et al., 2023). Two other methods, CoDeF (Ouyang et al., 2023) and TokenFlow (Geyer et al., 2023), both require per-video preparation. Specifically, CoDeF involves training a model for reconstructing the canonical image, while TokenFlow requires a 500-step DDIM inversion process to acquire the latent representation. Unlike all these methods requiring two-stage processing, our StreamV2V produces all the frames in a single stage. We continue to use xFormers (Lefaudeux et al., 2022) for fair comparison with existing methods. Our StreamV2V processes videos in 9 seconds with the 4-step LCM, making it significantly faster than current approaches. We note that while existing V2V methods use an A100-80GB GPU for runtime measurement, we only use an A100-40GB GPU. Despite this, StreamV2V remains significantly faster. Further acceleration is possible with fewer steps, though this may slightly affect performance. For more details, see Appendix A.4.

### 5.3.3 USER STUDY

We further conducted a user study to compare our method with five notable works. Following TokenFlow Geyer et al. (2023) and FlowVid Liang et al. (2023), we use the DAVIS dataset Pont-Tuset et al. (2017) with 67 video-prompt pairs. We adopt a Two-alternative Forced Choice (2AFC) protocol used in (Yang et al., 2023; Geyer et al., 2023), where participants are shown our result and a counterpart result, and are asked to identify which one has the best quality, considering both temporal consistency and text alignment. For each comparison, feedback was gathered from at least three different participants. The final win rates are shown in Figure 7. We find users significantly favor our StreamV2V over StreamDiffusion baseline (over 70% win rates) and CoDeF (over 80% win rates). It is important to note that while our method does not outperform the more advanced V2V methods such as Rerender, FlowVid, and TokenFlow, StreamV2V primarily aims to achieve *real-time video transfer*, rather than beating the state-of-the-art V2V methods, none of which support real-time processing and unlimited video length. Moreover, our StreamV2V can deal with half-minute long videos as showcased in Appendix A.6, while FlowVid and TokenFlow are limited to 4 seconds due to the memory limit (illustrated in Figure 2(c)). We have made all videos from the user study available on our project page for further examination. More details about the user study can be found in Appendix A.7.

### 5.4 ABLATION STUDY

We ablate several of our key designs in this section. The evaluation set contains 18 in-house videos with cartoon-style prompts.

### 5.4.1 EXTENDED SELF-ATTENTION (SA) AND FEATURE FUSION (FF)

As shown in Figure 8, the StreamDiffusion baseline, which doesn't have EA or FF, produces inconsistent outcomes. Noticeable flickering can be observed in the man's hands, hair, and the strips on his clothes. After introducing the EA (the third column), the consistency is much improved, with

the warp error decreased from 85.2 to 74.0. However, some artifacts persist in the man's hand. After further introducing FF (the last column), we have the most consistent result, with the lowest warp error of 73.4. We also isolate the effect of FF in the fourth column, we lower the warp error from 85.2 to 80.4 with the introduction of FF.

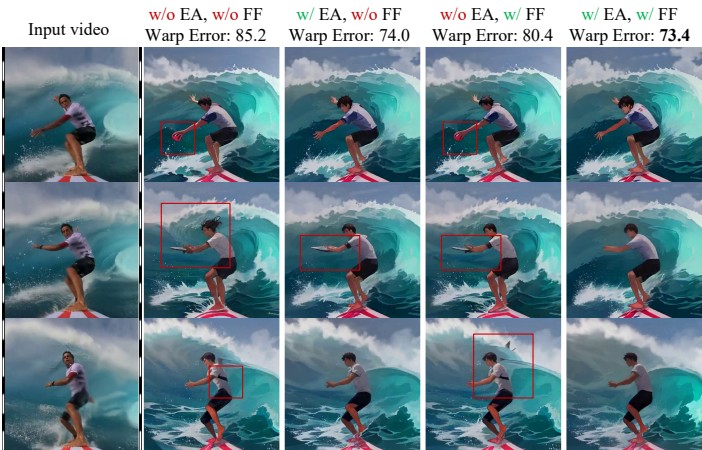

Figure 8: **Ablation on Extended self-Attention (EA) and Feature Fusion (FF).** Warp error is averaged within 18 videos.

Table 2: **Ablation on different feature banks.** No bank indicates the StreamDiffusion baseline. Queue represents the naive first-in-first-out queue. Our Dynamic Merging (DyMe) maintains a compact and informative bank.

| Bank type (size) | Time (ms) | Mem (GiB) | Warp error ↓ |
|---|---|---|---|
| no bank (0) | **60** | **3.68** | 85.2 |
| queue (1) | 70 | 4.03 | 76.4 |
| queue (2) | 79 | 4.17 | 74.8 |
| queue (4) | 100 | 4.77 | **72.4** |
| DyMe (1) | 75 | 4.07 | 73.4 |

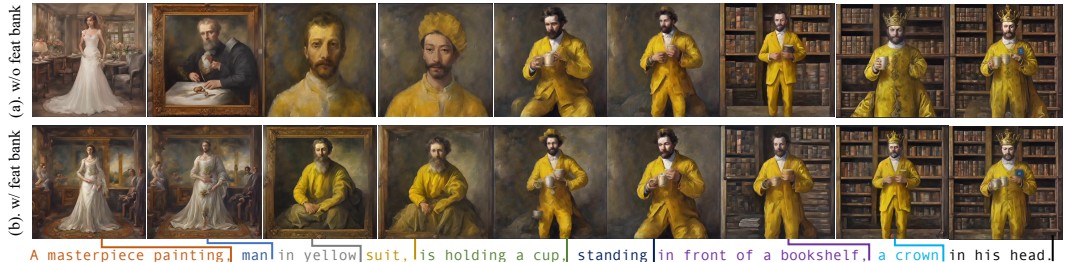

A masterpiece painting, man in yellow suit, is holding a cup, standing in front of a bookshelf, a crown in his head.

Figure 9: **Continuous text-to-image generation with feature bank.** The images from left to right are the intermediate generation as we type continuously. The caption of each column image is a segment of the prompt (separated by the line). (a) Per-frame LCM baseline has severe flickering even with slight one or two-word modification. (b) The feature bank provides a much smoother transition. We highly encourage readers to refer to video comparisons in our supplementary materials.

### 5.4.2 DYNAMIC MERGING (DYME) BANK

We study different feature banks in Table 2. The case of no bank, which represents the StreamDiffusion baseline, needs 60 ms to process a 512×512 frame on a single A100 GPU. However, its warp error is considerably high at 85.2. For queue-based banks, we test sizes of 1, 2, and 4. As expected, increasing bank size increases inference time but reduces warp error. Our dynamic merging (DyMe) bank achieves a running speed of 75 ms, faster than the queue bank of size 2, while maintaining a lower warp error. We also report the memory usage obtained via the nvidia-smi command. The introduced DyMe bank only adds a very small 10% GPU memory overhead when compared to StreamDiffusion.

### 5.5 EXTEND FEATURE BANK TO CONTINUOUS IMAGE GENERATION

The concept of the proposed feature bank, which links the current with the past, is expected to be broadly applicable beyond video-to-video. To demonstrate its versatility, we validate its effectiveness in a continuous text-to-image generation task.

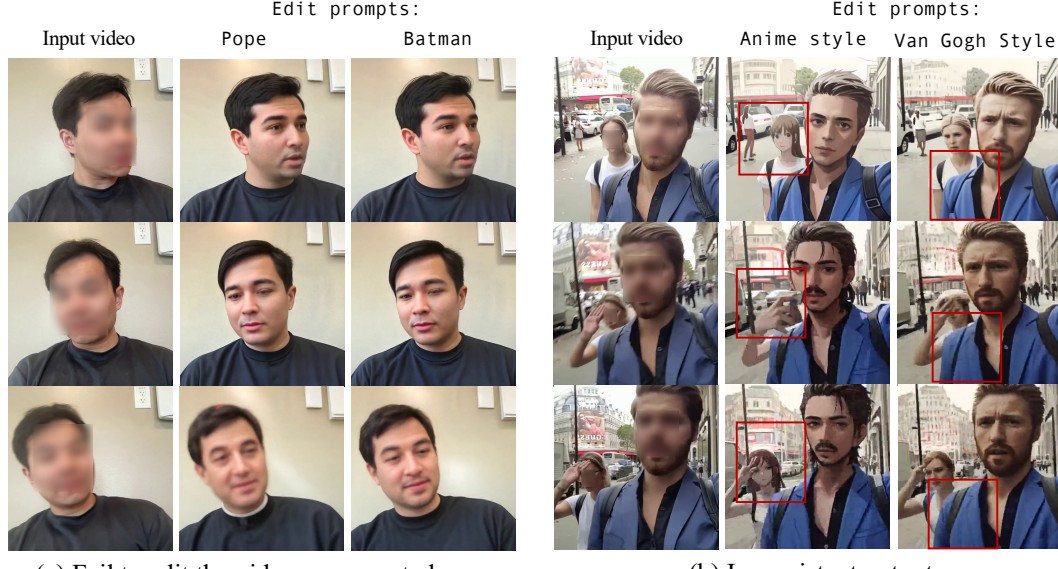

(a) Fail to edit the video as prompted  (b) Inconsistent output

Figure 10: **Limitations of StreamV2V.** (a). StreamV2V fails to alter the person within the input video into `Pope` or `Batman`. (b). StreamV2V can produce inconsistent output, as seen in the girl for `Anime style` and the backpack straps for `Van Gogh style`.

Continuous or real-time image generation allows users to produce images instantly when the prompt is being typed. However, most current frameworks generate each frame independently, resulting in flickering transitions between frames. Figure 9(a) illustrates per-frame generation in the LCM (Luo et al., 2023b). The images from left to right reflect the intermediate results as we type continuously. Even with slight one or two words, the layout of the image can shift dramatically. By incorporating a feature bank as in Figure 9(b), the model can reference previous frames, enabling smoother transitions.

## 5.6 LIMITATIONS

Despite achieving notable improvements, our model encounters certain limitations. First, our StreamV2V can fail to alter the video with the provided prompts. As shown in Figure 10(a), when we try to transfer the man to `Pope` or `Batman`, our model struggles to change the input video. This may be due to the use of SDEdit (Meng et al., 2021) as the primary image editing method, which has limited ability to significantly alter objects. We also find that increasing initial noise strength can improve editability but at the cost of introducing more pixel flickering. Another limitation is that StreamV2V can produce inconsistent output, especially for videos with rapid movements of the camera or the object. This is also the major reason why our StreamV2V still cannot match the performance of state-of-the-art FlowVid and TokenFlow. As demonstrated in Figure 10(b), when converting the video into `anime style`, the appearance of the girl in the background changes significantly. Similarly, when applying `Van Gogh style`, there are noticeable changes to the backpack straps of the man.

## 6 CONCLUSION

In this paper, we present StreamV2V: a diffusion model that can perform real-time video-to-video translation for streaming videos. The key to StreamV2V lies in a look-backward mechanism that enables the current frame to reason the past to ensure consistency. We propose a feature bank to store the intermediate features for past frames and reuse them in the current frame generation via extended self-attention and direct feature fusion. Furthermore, we propose dynamic merging, a method to make the bank compact and informative. Our evaluation shows that StreamV2V is an order of magnitude faster than existing video-to-video methods and can produce temporally consistent outputs.

ETHICS STATEMENT

Our research focuses on real-time video-to-video translation, aiming to unleash people's creativity for good applications. It is not designed to create harmful or misleading content, such as fraud or deepfake. However, like other related image/video generation models, it could still potentially be misused for deepfake humans. We strongly oppose any use cases that create fraudulent or harmful content using our model. Currently, the videos generated by our StreamV2V still contain human-perceivable artifacts, as shown in our video demos. StreamV2V demonstrates the possibility of using diffusion models to achieve real-time text-prompted video-to-video, but the overall editing performance is far from confusing people. However, when technology evolves, one day we may reach a point where generated videos are indistinguishable from real ones. We, as a community, need to investigate more research into detecting generated videos.

REPRODUCIBILITY STATEMENT

Our code is reproducible and can be implemented based on the method description in Section 4 as well as implementation details in Section 5.1. We have open-sourced our codes and models.

ACKNOWLEDGMENT

This research was supported in part by ONR Minerva program, NSF CCF Grant No. 2107085, iMAGiNE - the Intelligent Machine Engineering Consortium at UT Austin, and a UT Cockrell School of Engineering Doctoral Fellowship.

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

# A APPENDIX

## A.1 DIAGRAM OF STREAM BATCH

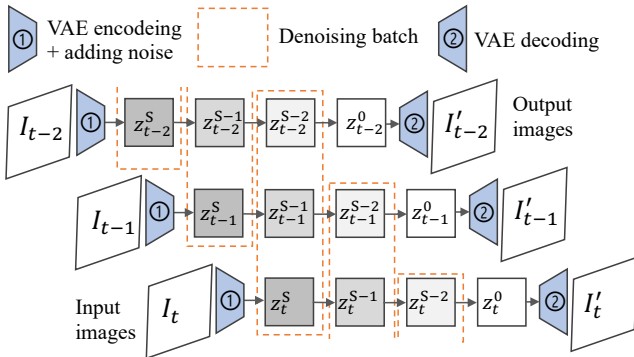

Figure 11: **Stream batch of StreamDiffusion.** The approach enables the processing of $S$ images with different denoising steps in a batch ($S$ is set to 3).

## A.2 HEATMAP VISUALIZATION OF EXTENDED SELF-ATTENTION

Given an anchor point (marked in red) in frame 4, we extract its query tensor from the last layer of the UNet up-block. We also obtain and store key tensors from the preceding frames 1, 2, and 3. We then calculate the dot product between the anchor query and stored key tensors. The most similar position is marked by the red point in frames 1, 2, and 3. This shows that the anchor point in frame 4 is highly correlated with the same goggle spot in earlier frames, indicating that extended self-attention functions as a weighted sum of similar areas across frames, thereby aligning the current frame with its past for improved consistency.

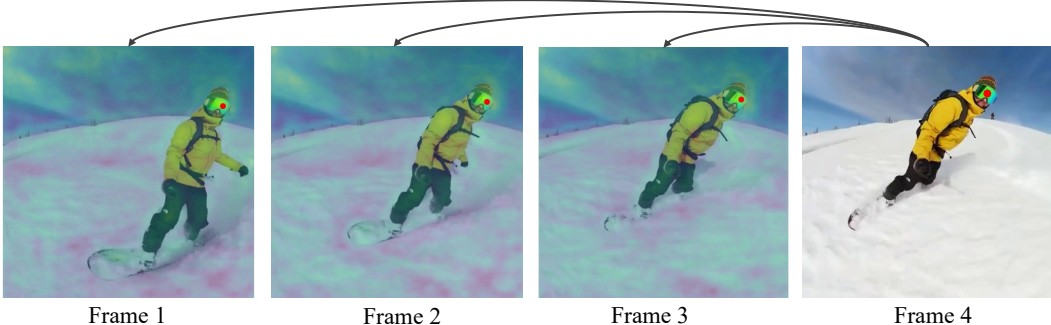

| Frame 1 | Frame 2 | Frame 3 | Frame 4 |

Figure 12: **Heatmap of extended self-attention.** The anchor point (marked in red) in frame 4 has a very high correlation with the same goggles spot in previous frames.

## A.3 DETAILS OF FEATURE FUSION

### A.3.1 FEATURE FUSION VISUALIZATION

To further verify the intuition of feature fusion, we visualize the output feature from the last layer of the UNet up-block. We begin by concatenating the features from all frames and applying principle component analysis (PCA) for visualization. The second row of Figure 13 shows the visualization of the first three principal components. We observe that similar concepts tend to appear in similar colors, indicating their similarity in feature space. For example, features corresponding to the goggles (blue box) and pants (red box) exhibit consistent coloring across frames, highlighting the potential of feature fusion to average them.

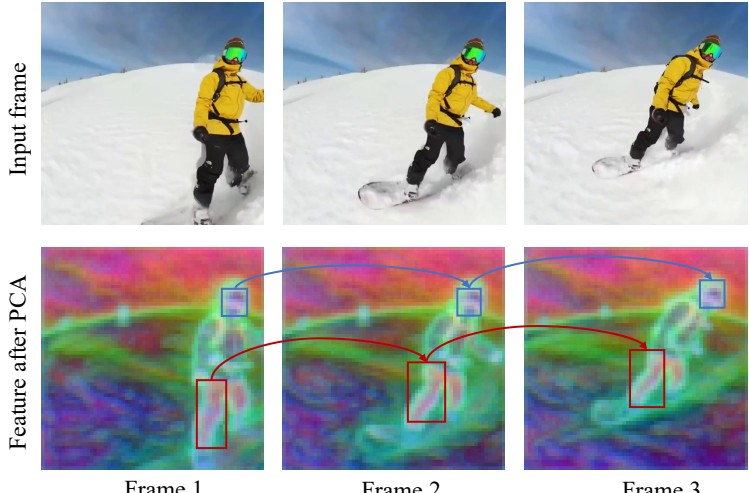

Figure 13: **Feature Fusion Illustration.** For input frames, PCA is applied to diffusion features extracted from all frames, and the first three principal components are visualized (second row). Features like the goggles (blue box) and pants (red box) show similar colors across frames, indicating consistent features.

### A.3.2    FEATURE FUSION POSITIONS

While feature fusion significantly enhances temporal consistency, it may result in blurry artifacts. As demonstrated in Figure 14, incorporating feature fusion in the layers makes the cat's face appear blurry. We believe that doing feature fusion at high-resolution features would average the features across frames, resulting in a diminishing of details. Thus, we suggest applying feature fusion only to low-resolution features, specifically within the middle block and the first two blocks of the upper block. As displayed in the right column of Figure 14, this approach preserves higher-quality details compared to using all layers.

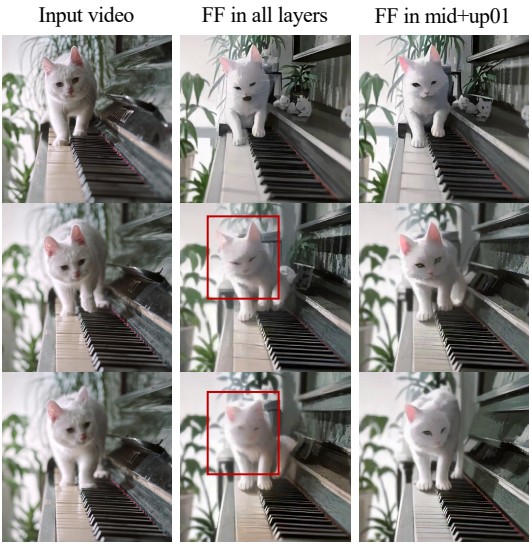

Figure 14: **Ablation on feature fusion (FF) layers.** Applying FF to all layers sometimes results in blurry artifacts, as seen in the cat's face. We propose to only apply FF to low-resolution layers, specifically the middle block (mid) and the first two blocks of the upper block (up01).

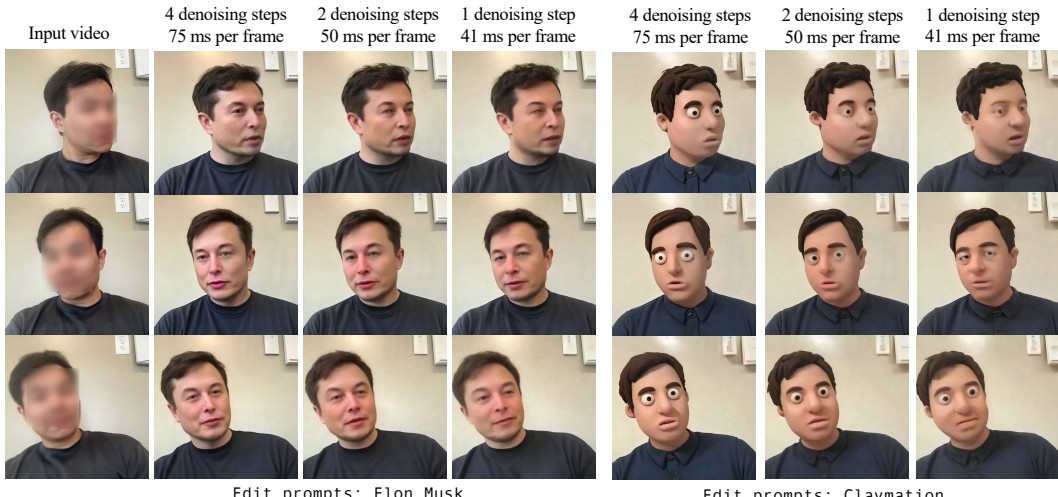

Figure 15: **Ablation on different denoising steps.** While using fewer denoising steps would accelerate the inference time for every frame, we do observe a certain level of quality drop if we use only 1 step.

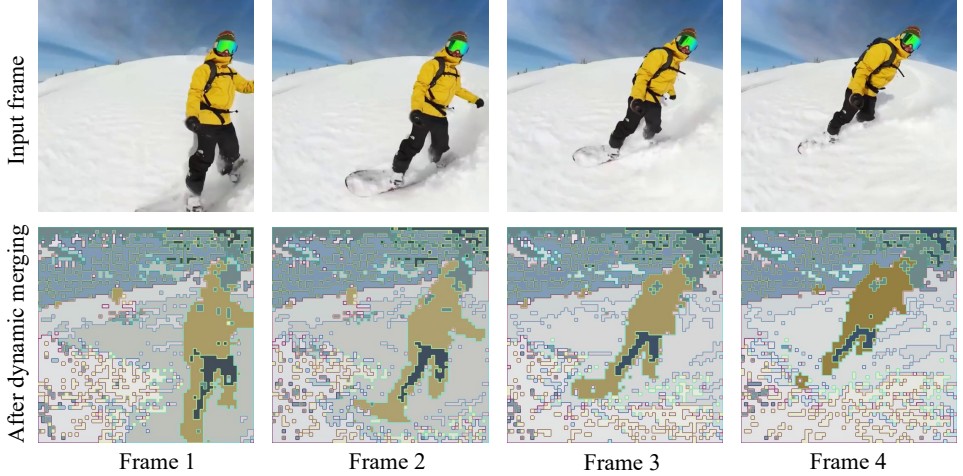

Figure 16: **Visualization of Dynamic Merging.** Similar patches across frames are merged, showcasing that the DyMe bank can maintain a compact yet informative feature bank.

## A.4 DETAILS OF DIFFUSION STEPS

As outlined in Section 5.1, our StreamV2V utilizes LCM (Luo et al., 2023b) as the accelerated diffusion model. LCM can perform denoising in 4, 2, or 1 steps, and we examine these variations in Figure 15. Reducing the number of steps decreases the inference time per frame from 75 ms to 41 ms. However, this also leads to a decline in video quality. Unless otherwise specified, we use the 4-step LCM to achieve optimal performance.

## A.5 MORE DETAILS OF DYNAMIC MERGING

### A.5.1 VISUALIZATION OF DYNAMIC MERGING

As discussed in Section 4.3, the Dynamic Merging (DyMe) bank merges redundant information across time. Figure 16 illustrates this effect. For better visualization, we applied the merging operation 10 times to the output features from the last layer of the UNet up-block, where merged features are

marked in the same color. We observe that key elements, such as the person, sky, and snow, are grouped consistently, demonstrating the effectiveness of dynamic merging.

### A.5.2  PSEUDO CODE OF DYNAMIC MERGING

```python
import torch
import torch.nn.functional as F

def dynamic_merge(current_frame, feature_bank):
    """
    Dynamic merging (DyMe) to create a compact feature bank.

    Args:
        current_frame (Tensor): Features of the current frame (shape: [N,
            D]).
        feature_bank (Tensor): Existing feature bank (shape: [N, D]).

    Returns:
        Tensor: Updated feature bank with dynamic merging.
    """
    # Step 1: Concatenate current frame features and feature bank
    all_features = torch.cat([current_frame, feature_bank], dim=0)  #
        Shape: [(2N, D]

    # Step 2: Randomly partition features into source (src) and
        destination (dst)
    num_features = all_features.size(0)
    permuted_indices = torch.randperm(num_features)
    src_indices = permuted_indices[:num_features // 2]
    dst_indices = permuted_indices[num_features // 2:]
    src_set = all_features[src_indices]  # Shape: [N, D]
    dst_set = all_features[dst_indices]  # Shape: [N, D]

    # Step 3: Find the most similar features between src and dst
    # Compute cosine similarity
    similarity_matrix = F.cosine_similarity(src_set.unsqueeze(1), dst_set
        .unsqueeze(0), dim=2)
    max_similarities, matched_indices = similarity_matrix.max(dim=1)

    # Step 4: Merge src into dst by averaging matched pairs
    for i, match_idx in enumerate(matched_indices):
        dst_set[match_idx] = (dst_set[match_idx] + src_set[i]) / 2

    updated_feature_bank = dst_set
    return updated_feature_bank
```

### A.6  MORE RESULTS ON LONG VIDEO

While most existing methods typically handle up to 4 seconds of video, StreamV2V can scale to arbitrary lengths thanks to the streaming processing and feature bank. As demonstrated in Figure 17, our approach handles a video exceeding 1000 frames (over 30 seconds) while maintaining consistent face swapping or style transfer throughout.

### A.7  MORE DETAILS OF USER STUDY

As described in Section 5.3.3, we conducted a user study to compare our method with existing approaches. Figure 18 illustrates the interface presented to each participant. The instructions given to users were as follows: The left/right video is labeled as A/B. You will be asked three questions to compare these two videos. If you perceive minimal differences between the videos, whether they are both good or bad, please select 'draw'.

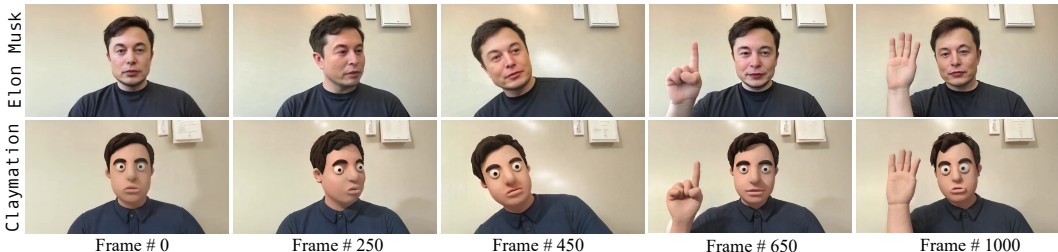

Figure 17: **Long video (> 1000 frames) generation.** Our StreamV2V can handle arbitrary length of videos without consistency degradation.

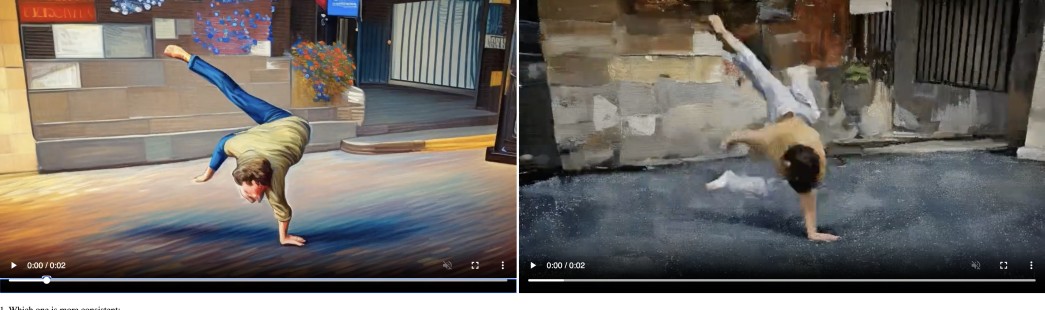

Figure 18: **User study Interface.** Each user is asked three questions about temporal consistency, prompt alignment and overall preference.

The participants are all college students aged 20-30. The ratio of male: female participants is roughly 2:1. The participants don't have hands-on research/engineering experience with generative videos but may have heard of the concept of generative AI. We find that user preference is very subjective and there are huge variances between users. We report the mean and standard deviation among at least three users in Table 2.

It is worth noting that, instead of providing numerical metrics for comparison, the study aims to offer a first-impression comparison between the two models. The results, whether indicating one model is better, worse, or on par with another, are generally consistent. This information can guide our decision-making, particularly when we need to balance speed and accuracy in model selection. In practice, we recommend using StreamV2V for applications where speed is a critical requirement, such as real-time webcam translation and draw rendering. For other uses, such as creating short video content, slower methods like FlowVid might yield better results.

Table 2: **Comparison of StreamV2V against various models**. We also report the mean and standard deviation (std) for each set of users.

| Model | StreamV2V wins (mean $\pm$ std) | Draws (mean $\pm$ std) | StreamV2V loses (mean $\pm$ std) |
|---|---|---|---|
| *vs.* StreamDiffusion | $71.1 \pm 10.2$ | $25.3 \pm 13.3$ | $3.6 \pm 5.2$ |
| *vs.* CoDeF | $81.3 \pm 20.1$ | $13.2 \pm 12.1$ | $5.5 \pm 9.3$ |
| *vs.* Rerender | $30.2 \pm 15.1$ | $31.4 \pm 7.8$ | $38.4 \pm 18.9$ |
| *vs.* FlowVid | $19.8 \pm 9.5$ | $21.3 \pm 8.5$ | $58.9 \pm 18.0$ |
| *vs.* TokenFlow | $19.3 \pm 9.0$ | $13.4 \pm 9.7$ | $67.3 \pm 13.4$ |

## A.8 More analysis and ablation studies

### A.8.1 Memory consumption with larger video resolution

We tested the memory footprint under different resolutions with one 24-GB A5000 GPU. As shown in Table 3, we can run StreamV2V up to a high resolution of $1536 \times 1536$. However, we did see a steady increase in the relative memory overhead of the DyMe bank. We conjecture the reason for this increase is that the baseline diffusion pipeline has specific optimization regarding memory usage, while our DyMe implementation doesn't. Some potential solutions are (1) storing features with key layers rather than all the layers; and (2) implementing memory optimization techniques for the DyMe data structure. We will consider these in our future work.

Table 3: **Memory Usage Comparison.** We report the memory usage (in GiB) of the baseline and StreamV2V across different resolutions, along with the relative memory overhead increase.

| Resolution | Mem of Baseline (GiB) | Mem of StreamV2V (GiB) | Relative Memory Overhead Increase |
|---|---|---|---|
| $512 \times 512$ | 4.87 | 5.34 | + 10% |
| $768 \times 768$ | 5.59 | 7.29 | + 30% |
| $1024 \times 1024$ | 6.62 | 9.86 | + 49% |
| $1536 \times 1536$ | 9.52 | 22.47 | + 136% |

### A.8.2 Similarity threshold of feature fusion

We ablate the effect of similarity threshold in feature fusion as in Table 4. The similarity threshold of 0.9 yields the lowest average warp error of 19 videos. We have worse results when the threshold is set to 1.0 (no features are fused) or 0.0 (all features are fused).

Table 4: **Effect of similarity threshold.** We analyze the impact of varying similarity thresholds on the average warp error of 19 videos. The best result, achieved with our default value of 0.9, is highlighted.

| Similarity Threshold | Warp Error $\downarrow$ |
|---|---|
| 0.0 | 74.0 |
| 0.8 | 73.5 |
| 0.9 (default) | **73.4** |
| 1.0 | 74.0 |

### A.8.3 Bank updating interval

We ablate the effect of bank updating intervals in Table 5. The interval of 8 brings us the best performance. During practice, we may want to adjust this value according to our videos: videos with larger motion may require a more frequent update.

Table 5: **Effect of bank updating interval.** We report the average warp error of 19 videos for different bank updating intervals. The best result and our method are highlighted.

| Bank Updating Interval | Warp Error $\downarrow$ |
|---|---|
| 1 | 78.1 |
| 2 | 75.3 |
| 4 (default) | 73.4 |
| 8 | 72.4 |
| 16 | 72.5 |

### A.8.4 Sampling strategy of DyME updating

Our DyMe bank needs to derive source (src) and destination (dst) sets from current and stored past features. We ablated three choices in Table 6. (1) Randomly sample (our default setting) ; (2) Uniformly grid sample: After concatenation, features at even locations are used as the source, and features at odd locations are used as the destination; and (3) Split sample: Use the current features as

Table 6: **Effect of sample strategy.** We evaluate warp error across different sampling strategies. The best result is highlighted.

| Sample Strategy | Warp Error ↓ |
|---|---|
| Random (default) | **73.4** |
| Uniform Grid | 73.7 |
| Split | 74.2 |

the destination, and the stored past features as the source. Split sampling has the worst warp errors, as it performs more like a queue bank, in which the current features dominate the bank. Compared to uniform grid sampling, random sampling proved more general and achieved the lowest warp error. Thus, random sampling was chosen as the default strategy.

