# OpenReview forum: "Looking Backward: Streaming Video-to-Video Translation with Feature Banks"
_ICLR.cc/2025/Conference — ICLR 2025 Poster_

### Official Review · Reviewer_bNTf · 2024-10-28

**Soundness:** 3
**Presentation:** 3
**Contribution:** 4
**Rating:** 6
**Confidence:** 3

**Summary:**

This paper presents a study on real-time video-to-video (V2V) translation for streaming input using diffusion models. The authors identify limitations in batch-processing V2V methods, which are constrained by GPU memory and unsuitable for continuous, long-duration videos. In response, they propose StreamV2V, a diffusion model framework that integrates a feature bank, allowing it to maintain frame-to-frame consistency by leveraging stored features from past frames. This architecture enables StreamV2V to operate at 20 FPS on a single A100 GPU, significantly faster than existing V2V methods, and to support streaming inputs of indefinite length. Experimental results confirm that StreamV2V achieves competitive consistency and temporal stability, with favorable feedback from user studies compared to other V2V models. This paper is well-organized, and the methods and results are clearly presented.

**Strengths:**

1.This paper addresses a critical need in real-time video-to-video (V2V) translation, especially for streaming applications, which is highly relevant to the field.
2.The paper is well-organized, with clear explanations of both the limitations in existing methods and the motivations behind the proposed backward-looking feature bank approach. The authors’ solution is explained in a logical and accessible manner, supporting reader understanding.

**Weaknesses:**

1.The model sometimes struggles to maintain temporal consistency when handling videos with significant camera or object movement, which may impact the overall quality.
2.It may have limited capacity for major visual transformations, such as altering complex object characteristics, which may restrict its applicability in highly dynamic editing tasks.

**Questions:**

I would like to understand the rationale behind first concatenating, then randomly selecting features for matching and updating during the feature bank update process. Were other methods tested in the experiments, such as retaining a specific proportion of past and current features for matching and updating? Would the performance change if most of the randomly selected features came from either past frames or the current frame?

---

> ### Author Response · Authors · 2024-11-26
> **Response to Reviewer bNTf**
>
> We appreciate the reviewer recognizing our StreamV2V addressing the critical need for real-time video-to-video, the paper is clearly organized and easy to understand. We are happy to address the reviewer’s questions.
>
> **Q1:** The model sometimes struggles to maintain temporal consistency when handling videos with significant camera or object movement, which may impact the overall quality
>
> **A1:** We acknowledge this limitation, as it is a key reason why StreamV2V does not yet match the performance of state-of-the-art methods. Significant motion reduces the effectiveness of our training-free extended self-attention and feature fusion, as features change too drastically. A potential solution is to train the memory attention mechanism, as explored in concurrent work like SAM-2 [4], to better leverage the memory bank through a learned temporal module. However, this is beyond the scope of StreamV2V and is identified as future work. We do note StreamV2V is the only existing real-time streaming approach, so a head-to-head comparison with other non-streaming/batch processing approaches may not be appropriate.
>
> **Q2:** It may have limited capacity for major visual transformations, such as altering complex object characteristics, which may restrict its applicability in highly dynamic editing tasks.
>
> **A2:** We use SDEdit  as the primary image editing method, which has limited ability to significantly alter objects. Even though increasing the initial noise strength can improve editability, we find it harder to maintain temporal consistency if we have larger edits (such as changing a heavy-set man to a slim woman).
>
> **Q3:** I would like to understand the rationale behind first concatenating, then randomly selecting features for matching and updating during the feature bank update process. Were other methods tested in the experiments, such as retaining a specific proportion of past and current features for matching and updating? Would the performance change if most of the randomly selected features came from either past frames or the current frame?
>
> **A3:** This is obtained from an experimental ablation study. We ablated three choices to derive source (src) and destination (dst) sets as follows: (1) Randomly sample (our default setting) ; (2) Uniformly grid sample: After concatenation, features at even locations are used as the source, and features at odd locations are used as the destination; and  (3) Split sample: Use the current features as the destination, and the stored past features as the source. The results are included below:
>
> | Sample strategy  | Warp error |
> |------------------|------------|
> | Random           | 73.4       |
> | Uniform grid     | 73.7       |
> | Split            | 74.2       |
>
>
> Split sampling has the worst warp errors, as it performs more like a queue bank, in which the current features dominate the bank. Compared to uniform grid sampling, random sampling proved more general and achieved the lowest warp error. Thus, random sampling was chosen as the default strategy.

---

### Official Review · Reviewer_wGbG · 2024-11-02

**Soundness:** 3
**Presentation:** 3
**Contribution:** 3
**Rating:** 8
**Confidence:** 3

**Summary:**

This paper introduces StreamV2V, a real-time video-to-video translation model that uses a feature bank and backward-looking mechanism to ensure temporal consistency. StreamV2V achieves significantly faster speeds than existing methods while maintaining consistent outputs.

**Strengths:**

1. The paper proposes a novel approach to streaming video-to-video translation, introducing a feature bank mechanism to address temporal consistency, enabling each frame in the video stream to reference features from past frames in an innovative manner.

2. This method achieves efficient real-time processing without model fine-tuning, outperforming many comparable methods in experimental results.

3. StreamV2V requires no training or fine-tuning and seamlessly integrates with existing image diffusion models, demonstrating the potential for a wide range of video editing tasks.

4. The paper includes extensive quantitative experiments and user studies, validating its performance.

5. This paper is well written and easy to follow. The supplementary materials also include comprehensive experimental details, content, and discussions, demonstrating the effectiveness of the proposed method.

**Weaknesses:**

1. Does DyMe Bank still have advantages in video quality compared to queue-based banks when dealing with longer videos, more complex video content, or video motion?

**Questions:**

See Weaknesses.

---

> ### Author Response · Authors · 2024-11-26
> **Response to Reviewer wGbG**
>
> We appreciate the reviewer’s recognition of our novel feature bank mechanism, efficient real-time performance, and seamless integration with diffusion models. We are also pleased that the experiments and supplementary materials were found valuable. We are happy to address the reviewer’s concerns.
>
> **Q1:** Does DyMe Bank still have advantages in video quality compared to queue-based banks when dealing with longer videos, more complex video content, or video motion?
>
> **A1:** When we ablated the effects of bank type and size in Section 5.4.2, our evaluation set contains several long videos up to 10 seconds with dynamic motion. We find that DyMe generally works better compared to the same-size queue bank both in eye-ball observation or lower warp error. We showcased the long-video generation with DyMe in Appendix A.6 where we can find our DyMe bank can keep a good consistency up to 30 seconds.

---

> > ### Comment · Reviewer_wGbG · 2024-12-02
> >
> > I appreciate the authors’ response and will maintain my score.

---

### Official Review · Reviewer_DwFq · 2024-11-03

**Soundness:** 3
**Presentation:** 3
**Contribution:** 3
**Rating:** 8
**Confidence:** 3

**Summary:**

The paper presents StreamV2V, a diffusion model for real-time, streaming video-to-video (V2V) translation using user prompts. It achieves efficiency by maintaining a continually updated feature bank of past frames, which it references to enhance incoming frames without needing fine-tuning. StreamV2V seamlessly integrates with image diffusion models and operates at 20 FPS on an A100 GPU, making it significantly faster than competing models. User studies and quantitative metrics validate StreamV2V's strong temporal coherence and adaptability.

**Strengths:**

1. this paper addresses the challenge of real-time video-to-video translation via the streamV2V framework with backward-looking and training-free techniques. Since streamV2V is training-free and fine-tuning-free compared to other video-to-video techniques, the proposed approach is feasible to be deployed to practical applications.

2. the streamV2V framework enhances coherence between adjacent translated video frames through the extended self-attention layer and a proposed feature fusion approach for intermediate features of the current and the past frames. Due to the extended self-attention layer and the feature fusion approach, streamV2V effectively reuses the intermediate features from the past video frames to generate coherent translated video frameworks between adjacent frames.

3. the streamV2V stores projected keys, projected values, and outputs of intermediate features of the past video frames through a feature bank with a dynamic merging mechanism. With the dynamic feature bank, the streamV2V framework trivially reuses historical features to align the outputs of intermediate blocks of the current and the past frames.

**Weaknesses:**

1. the proposed streamV2V framework fails to perform video-to-video translation for high-quality video generation in some scenarios while a prompt is provided.

2. the proposed streamV2V framework fails to perform video-to-video translation when input video includes rapid motion movements.

3. to test the generalization of the proposed framework, the manuscript misses experimentation comparisons for variations of difusion models.

**Questions:**

1. how does the proposed framework still outperform other video-to-video translation techniques for variations of diffusion models? Do different models with the proposed framework still outperform other video-to-video translation techniques?

2. what does the running-time comparison look like on CPU Inferences rather than GPU inferences?

---

> ### Author Response · Authors · 2024-11-26
> **Response to Reviewer DwFq**
>
> We appreciate the reviewer’s feedback highlighting our StreamV2V as feasible to be deployed to practical applications. We are happy to address the reviewer’s concerns.
>
> **Q1:**  the proposed streamV2V framework fails to perform video-to-video translation for high-quality video generation in some scenarios while a prompt is provided.
>
> **A1:** Our StreamV2V aims to improve the temporal consistency for real-time video-to-video. The prompt following capability is determined by the base diffusion model SD 1.5 and the image editing method SDEdit. In our failures shown in Figure 10 (a), we find it is caused by SDEdit's failure to edit the image accordingly. A potential solution is to use more advanced base models and image editing methods.
>
> **Q2:** the proposed streamV2V framework fails to perform video-to-video translation when input video includes rapid motion movements.
>
> **A2:** We acknowledge this limitation because this is the major reason why our StreamV2V still cannot match the performance of state-of-the-art. Rapid motion reduces the effectiveness of our training-free extended self-attention and feature fusion, as features change too drastically.  A potential solution is to train the memory attention mechanism, as in concurrent work like SAM-2 [1], to better utilize the memory bank through a learned temporal module. However, this is beyond the scope of StreamV2V and is highlighted as future work.
>
> **Q3:** to test the generalization of the proposed framework, the manuscript misses experimentation comparisons for variations of difusion models.
>
> **A3:** We chose SD 1.5 as our major diffusion model because it is widely used by other video-to-video methods. We are adding hooks to diffusion blocks to store features and reuse stored features for new frames, which is extendable to other diffusion variations. We even validate the generalization by showing that the feature bank could provide a smoother transition when doing continuous text-to-image as detailed in Section 5.5.
>
> **Q4:** how does the proposed framework still outperform other video-to-video translation techniques for variations of diffusion models? Do different models with the proposed framework still outperform other video-to-video translation techniques?
>
> **A4:** We tested only SD 1.5, as other V2V methods also used SD 1.5 for fair comparison. We conjecture that if all methods, including StreamV2V and other V2V techniques, use the same diffusion models, the relative ranking among the methods is likely to remain consistent.
>
> **Q5:** what does the running-time comparison look like on CPU Inferences rather than GPU inferences?
>
> **A5:** Our StreamV2V runs around 6 seconds per image (12 mins for 120 frames) on AMD EPYC 7742 64-Core CPU Processor. Using the same CPU, TokenFlow took 8 hours to process the same video.
>
> [1] Ravi, Nikhila, et al. "Sam 2: Segment anything in images and videos." arXiv preprint arXiv:2408.00714 (2024).

---

> > ### Comment · Reviewer_DwFq · 2024-11-26
> >
> > Since the rebuttal addresses all concerns I had, I decide to raise my rating.

---

### Official Review · Reviewer_7fxd · 2024-11-03

**Soundness:** 3
**Presentation:** 3
**Contribution:** 3
**Rating:** 6
**Confidence:** 4

**Summary:**

The paper describes a video-to-video translation approach  that generates video frames in a streaming fashion. The method is based on StreamDiffusion for batch processing of frames and uses SDedit for generating each frame. The method needs no training or fine-tuning can be used together with any image diffusion model. Although the results are superior to some recent video translation models, they are inferior to the results of TokenFlow and FlowVid. However the method excels in its inference time.

**Strengths:**

The paper has the following novel contributions: 1) videos can be translated in a streaming fashion without the need to process long sequences of video frames, 2) The past information of previous frames is stored in a feature bank and used during the generation of current frame without the need to fine-tune the existing image generation model, 3) Extended self-attention and feature fusion techniques are introduced to provide temporal consistency between consecutive frames. The most important contributions are the streaming video generation capability and the ability to use the method as an add-on with any image generation model without any training or fine-tuning.

**Weaknesses:**

The proposed method is fast and efficient, but it lags behind state-of-the-art  video translation models in CLIP score and subjective evaluations. The level of novelty is marginal: the paper does not propose a new video translation architecture, but merely develops an efficient feature storage, update and fusion strategy for temporal consistency. The overall approach is based on the existing work of StreamDiffusion and SDEdit.

**Questions:**

The authors claim that the method can be used as an add-on with any image generation model without any training or fine-tuning. This claim needs further elaboration. How is it that the method can be integrated to any translation model without any fine-tuning? How do the authors decide which features to store in the feature bank and which features to use in feature fusion? Could some fine-tuning improve the performance of the overall model? If not, why? If so, why haven't the authors done that so that overall performance could be improved?

---

> ### Author Response · Authors · 2024-11-26
> **Response to Reviewer 7fxd**
>
> We appreciate the reviewer’s feedback highlighting our StreamV2V as fast, efficient, and flexible to be used as an add-on with any image generation model. We are happy to address the reviewer’s concerns.
>
> **Q1:** the paper does not propose a new video translation architecture, but merely develops an efficient feature storage, update, and fusion strategy for temporal consistency.
>
> **A1:**  While StreamV2V builds on existing methods like StreamDiffusion and SDEdit, the use of a feature bank in streaming video-to-video translation is novel. Additionally, to the best of our knowledge, ours is among the first approaches to address real-time video-to-video translation.
>
> **Q2:** How is it that the method can be integrated to any translation model without any fine-tuning?
>
> **A2:** Implementation-wise, we are adding hooks to diffusion blocks to store features and reuse stored features for new frames. We can use different community variants of a base diffusion model, such as LoRAs from Civita, without changing any codes. In our camera demo on our supplementary website, we are using different LoRAs to achieve better stylization. Moreover, the idea of the feature bank can be generalized to text-to-image generation to empower a smoother transition between frames, which is detailed in Section 5.5
>
> **Q3:** How do the authors decide which features to store in the feature bank and which features to use in feature fusion?
>
> **A3:** For extended self-attention, we store features from all layers, as skipping some layers leads to performance degradation. For feature fusion, we focus on low-resolution features, specifically within the middle block and the first two blocks of the upper block. Applying feature fusion to all layers can sometimes result in blurry artifacts (details are provided in Appendix A.3.2).
>
> **Q4:** Could some fine-tuning improve the performance of the overall model? If not, why? If so, why haven't the authors done that so that overall performance could be improved?
>
> **A4:**  We believe proper video fine-tuning could enhance overall performance. For example, an extension of StreamV2V could involve training the memory attention mechanism, as explored in concurrent work like SAM-2 [1], to better leverage the memory bank through a learned temporal module. However, this paper focuses on proposing an efficient training-free solution for real-time video-to-video translation. While training-based approaches are indeed promising, they fall outside the scope of our current work
>
> [1] Ravi, Nikhila, et al. "Sam 2: Segment anything in images and videos." arXiv preprint arXiv:2408.00714 (2024).

---

> > ### Comment · Reviewer_7fxd · 2024-11-29
> >
> > The authors have mostly answered my comments. Therefore I decided to raise the overall score of the paper.

---

### Official Review · Reviewer_KRkM · 2024-11-04

**Soundness:** 3
**Presentation:** 3
**Contribution:** 3
**Rating:** 6
**Confidence:** 4

**Summary:**

This paper presents a real-time zero-shot method for streaming video-to-video translation based on diffusion models. The key innovation is the adoption and maintenance of a dynamic feature bank which significantly increases the cross-frame consistency in the resultant videos. Although Steam Diffusion is existing work and there are a huge amount of memory updating algorithms in the field of video understanding and visual tracking, this reviewer appreciates the soundness and practical value of the proposed approach.

**Strengths:**

This work is well-motivated and addresses an important problem. The method proposed is technically sound and practically feasible. The method achieves real-time V2V translation speed with a translation quality on par with the SOTA methods. The experiments, especially the ablation studies, are convincing enough to show the system-level contribution of the work.

**Weaknesses:**

There are still some artifacts in the result videos. This reviewer wonders whether a training-free method could eventually achieve a satisfactory performance. This will need some discussions.
Component wise, the innovation of this work is not very high. But as stated above, this reviewer appreciates the system-level contributions of the work.
In quantitative metrics comparison, the CLIP score is not quite differentiating.

**Questions:**

Refer to the first question in weakness.

**Details Of Ethics Concerns:**

The examples shown in the paper and supplementary material already raise some concern about privacy and deep fake.

---

> ### Author Response · Authors · 2024-11-26
> **Response to Reviewer KRkM**
>
> We appreciate the reviewer’s feedback highlighting our StreamV2V as well-motivated, technically sound and practical feasible. We are happy to address the reviewer’s concerns.
>
> **Q1:** There are still some artifacts in the result videos. This reviewer wonders whether a training-free method could eventually achieve a satisfactory performance. This will need some discussions.
>
> **A1:** While our training-free method provides flexibility, it lags behind training-based methods like FlowVid. We also agree that training-based methods are more likely to have a higher ceiling regarding the performance. A potential extension of StreamV2V is to train the memory attention as in concurrent SAM-2 [1]. Intuitively, we want to learn a temporal module to exploit the memory bank better. We point to this extension as our future work.
>
> **Q2:** In quantitative metrics comparison, the CLIP score is not quite differentiating.
>
> **A2:** CLIP scores lack differentiation because CLIP compresses an image into a single feature vector, which limits its ability to capture nuances and details. This is consistent with reports from other papers, such as Rerender and Pix2Video. We’d like to note the quantitative metrics, such as CLIP scores, are reference-only and sometimes don’t indicate the quality of the generated videos. User studies provide a more accurate measure of real human preferences.
>
> [1] Ravi, Nikhila, et al. "Sam 2: Segment anything in images and videos." arXiv preprint arXiv:2408.00714 (2024).

---

> > ### Comment · Reviewer_KRkM · 2024-12-02
> > **I will keep my positive score**
> >
> > I appreciate the authors’ response and will keep my score.

---

### Official Review · Reviewer_geFz · 2024-11-05

**Soundness:** 4
**Presentation:** 4
**Contribution:** 3
**Rating:** 6
**Confidence:** 3

**Summary:**

StreamV2V is an online approach (as opposed to batch / offline) for generating continuous video-to-video translation output without limits to video length and runs at 20 fps on a single A100 GPU.  The core idea behind StreamV2V is a feature bank from past frames that is retrieved and fused to ensure temporal coherence and consistency in frame generation.

**Strengths:**

- The core idea of the paper makes sense and is clearly presented.
- The method achieves good results, both in terms of computational time (50 ms per frame) and quality (using objective metrics and subjective user studies).
- Limitations and weaknesses against SOTA methods are presented.

**Weaknesses:**

- Some magic constants are reported without explanation (e.g., threshold of 0.90, max number of frames of 2, update feature bank every 4 frames).  How sensitive is your methods to these parameters?  What are the impact of higher or lower values?
- The user study is presented without the necessary details.  How many participants? Age group? Gender? Prior experience with generative videos?  The authors should refer to papers reporting user studies to see what is the norm of doing so (e.g., see papers from CHI)
- The choice of SDEdit (which is 3 years old) should be explained -- why not newer methods?

**Questions:**

Please clarify the points mentioned in weaknesses above.  Particularly:

- How are the magic constants determined and have you studied the impact?  If so, what did you find?

- Please provide the details of the user study.

- Please articulate the design choices, particularly SDEdit.

---

> ### Author Response · Authors · 2024-11-26
> **Response to Reviewer geFz**
>
> We appreciate the reviewer’s feedback, recognizing that StreamV2V's core idea is clear and makes sense, achieves good results in both computational time and quality, and presents its limitations and weaknesses against SOTA methods. We are happy to address the reviewer’s questions.
>
> **Q1:** Some magic constants are reported without explanation (e.g., threshold of 0.90, max number of frames of 2, update feature bank every 4 frames). How sensitive is your methods to these parameters? What are the impact of higher or lower values?
>
> **A1:** These values are determined by detailed ablation studies. As shown in the following table, the similarity threshold of 0.9 yields the lowest average warp error of 19 videos. When the threshold is set to 1.0 (no features are fused) or 0.0 (all features are fused), we have worse results.
>
> | Similarity threshold | Warp error |
> |----------------------|------------------------------------|
> | 0.0                  | 74.0                               |
> | 0.8                  | 73.5                               |
> | 0.9 (ours)           | 73.4                               |
> | 1.0                  | 74.0                               |
>
> The number of frames of 2 (L259-260) is only for demonstration of dynamic merging. We ablate the size of the queue bank in Table 2. Our dynamic merging bank is using size 1.
>
> We also ablated updating intervals in the following table. We find that the default setting 4 is not optimal. Increasing the interval to 8 further lowers the warp error.
>
> | Bank frame interval | Warp error  |
> |---------------------|------------------------------------|
> | 1                   | 78.1                              |
> | 2                   | 75.3                              |
> | 4 (ours)            | 73.4                              |
> | 8                   | 72.4                              |
> | 16                  | 72.5                              |
>
> We will include all these ablations in our updated version.
>
>
> **Q2:** The user study is presented without the necessary details. How many participants? Age group? Gender? Prior experience with generative videos? The authors should refer to papers reporting user studies to see what is the norm of doing so (e.g., see papers from CHI)
>
> **A2:** For each set of comparisons (StreamV2V vs. one other method), we collected feedback from at least three different participants. The participants are all college students aged 20-30. The ratio of male:female participants is roughly 2:1. The participants don’t have hands-on research/engineering experience with generative videos but may have heard of the concept of generative AI. As we detailed in Appendix A.7, our user study aims to offer a first-impression comparison between the two models, rather than providing a precise numerical comparison. The results, whether indicating one model is better, worse, or on par with another, are generally consistent across users.
>
> **Q3:** The choice of SDEdit (which is 3 years old) should be explained -- why not newer methods?
>
> **A3:** We selected SDEdit because StreamV2V focuses on real-time video-to-video translation, and SDEdit offers high efficiency. Newer methods like PnP require additional DDIM inversion, while methods such as ControlNet and T2I-Adapter rely on extra modules to process images, which adds computational overhead. We validate the generalization capability of the feature bank (our core contribution) with continuous text-to-image generation in Section 5.5. We believe the concept of a feature bank can be generalized to other image editing methods, too.

---

> > ### Comment · Reviewer_geFz · 2024-11-26
> >
> > Thank you for clarifying the details above and for conducting/reporting additional ablation results.
> >
> > My concern is with the reported subjective evaluation.
> >
> > Having (at least) 3 participants is not enough -- particularly since you mentioned that there is a huge variance among users.  I would expect 15 or more for a statistically significant result.
> >
> > Furthermore, it is a common practice to report the exact sample size.  Saying "at least" is too vague.  Similarly, "roughly" 2:1 is not precise enough in a scientific context.
> >
> > I would like to iterate that there are norms in how researchers conduct and report user studies, and the authors should pay more attention to them to bring rigor to the reported evaluation results.

---

### Official Review · Reviewer_cUnp · 2024-11-05

**Soundness:** 3
**Presentation:** 3
**Contribution:** 3
**Rating:** 6
**Confidence:** 4

**Summary:**

This paper presents StreamV2V, a real-time streaming video-to-video (V2V) translation method leveraging a backward-looking feature bank to maintain temporal consistency across video frames. Unlike traditional batch-processing approaches, StreamV2V processes incoming video frames in a streaming fashion, which allows it to handle an unlimited number of frames without fine-tuning. The method introduces a feature bank that stores intermediate features from past frames, extended self-attention, and direct feature fusion to improve temporal consistency. Furthermore, a dynamic feature bank updating strategy is implemented to reduce redundancy and optimize efficiency. The experimental results show that StreamV2V can achieve real-time processing at 20 FPS on a single A100 GPU, significantly outperforming existing methods like FlowVid, CoDeF, Rerender, and TokenFlow in terms of speed, with competitive consistency performance.

**Strengths:**

- The motivation behind StreamV2V is clear and addresses an essential limitation in existing video-to-video methods that require batch processing and are constrained by GPU memory.
- The implementation is designed for practical efficiency, with a clear description of feature fusion and dynamic feature bank updates, ensuring minimal memory overhead and ease of replication.
- The method provides flexible control over generating different styles without requiring model fine-tuning, allowing users to adapt the output to various visual aesthetics, which makes it suitable for creative applications.

**Weaknesses:**

- A significant concern is that using the feature bank to generate higher resolution could lead to a drastic increase in memory usage.
- The DyMe (Dynamic Merging) mechanism lacks detailed pseudocode to illustrate the workflow, which would help in understanding and replicating the process.

**Questions:**

- What is the size of the feature bank used in the experiments? Does the size of the feature bank affect the performance of StreamV2V?
- Can the proposed method be applied to previous text-to-video approaches to improve the performance of long video generation?

---

> ### Author Response · Authors · 2024-11-26
> **Response to Reviewer cUnp**
>
> We appreciate the reviewer’s feedback in recognizing our StreamV2V as clearly-motivated, efficient in memory overhead, and flexible to generate different styles without fine-tuning. We are happy to address the reviewer’s questions.
>
> **Q1:** A significant concern is that using the feature bank to generate higher resolution could lead to a drastic increase in memory usage.
>
> **A1:**  We tested the memory footprint under different resolutions with one 24-GB A5000 GPU. As shown in the following table, we can run StreamV2V up to a high resolution of 1536 × 1536. However, we did see a steady increase in the relative memory overhead of the DyMe bank. We conjecture the reason for this increase is that the baseline diffusion pipeline has specific optimization regarding memory usage, while our DyMe implementation doesn’t. Some potential solutions are (1) storing features with key layers rather than all the layers; and (2) implementing memory optimization techniques for the DyMe data structure. We will consider these in our future work.
>
> | Resolution     | Mem of Baseline (GiB) | Mem of StreamV2V (GiB) | Relative memory overhead increase |
> |----------------|---------------------|------------------------------|-----------------------------------|
> | 512 × 512      | 4.87               | 5.34                        |  + 10%                              |
> | 768 × 768      | 5.59               | 7.29                        |  + 30%                              |
> | 1024 × 1024    | 6.62               | 9.86                        |  + 49%                              |
> | 1536 × 1536    | 9.52               | 22.47                       | + 136%                            |
>
> **Q2:** The DyMe (Dynamic Merging) mechanism lacks detailed pseudocode to illustrate the workflow, which would help in understanding and replicating the process.
>
> **A2:** We provide the pseudo codes with Pytorch as follows. We will also release all the codes upon publication.
>
> ```python
> import torch
> import torch.nn.functional as F
>
> def dynamic_merge(current_frame, feature_bank):
>     """
>     Dynamic merging (DyMe) to create a compact feature bank.
>
>     Args:
>         current_frame (Tensor): Features of the current frame (shape: [N, D]).
>         feature_bank (Tensor): Existing feature bank (shape: [N, D]).
>
>     Returns:
>         Tensor: Updated feature bank with dynamic merging.
>     """
>     # Step 1: Concatenate current frame features and feature bank
>     all_features = torch.cat([current_frame, feature_bank], dim=0)  # Shape: [(2N, D]
>
>     # Step 2: Randomly partition features into source (src) and destination (dst)
>     num_features = all_features.size(0)
>     permuted_indices = torch.randperm(num_features)
>     src_indices = permuted_indices[:num_features // 2]
>     dst_indices = permuted_indices[num_features // 2:]
>     src_set = all_features[src_indices]  # Shape: [N, D]
>     dst_set = all_features[dst_indices]  # Shape: [N, D]
>
>     # Step 3: Find the most similar features between src and dst
>     # Compute cosine similarity
>     similarity_matrix = F.cosine_similarity(src_set.unsqueeze(1), dst_set.unsqueeze(0), dim=2)
>     max_similarities, matched_indices = similarity_matrix.max(dim=1)
>
>     # Step 4: Merge src into dst by averaging matched pairs
>     for i, match_idx in enumerate(matched_indices):
>         dst_set[match_idx] = (dst_set[match_idx] + src_set[i]) / 2
>
>     updated_feature_bank = dst_set
>     return updated_feature_bank
> ```
>
> **Q3:** What is the size of the feature bank used in the experiments? Does the size of the feature bank affect the performance of StreamV2V?
>
> **A3:** Our DyMe bank has a size of 1, meaning we store the intermediate features of a single image. We ablated increased bank size and performance of a naive queue bank in Table 2. We believe the same conclusion will also hold for DyMe banks: increased bank size results in better consistency but comes at the cost of slower inference speed and higher memory usage (noted in Table 2)
>
> **Q4:** Can the proposed method be applied to previous text-to-video approaches to improve the performance of long video generation?
>
> **A4:** Similar to video-to-video, text-to-video models also face memory constraints when generating long videos. Researchers are using autoregressive processing (akin to our proposed streaming process) instead of batch processing to handle long video generation [1][2]. Moreover, as illustrated in the continuous text-to-image experiment in Section 5.5 and Figure 9, the proposed feature bank can reason the current to the past, which will help the long video to maintain a better consistency across time. Extending our method to text-to-video generation is a promising future work.
>
> [1] Henschel, Roberto, et al. "Streamingt2v: Consistent, dynamic, and extendable long video generation from text." arXiv preprint arXiv:2403.14773 (2024).
>
> [2] Xie, Desai, et al. "Progressive autoregressive video diffusion models." arXiv preprint arXiv:2410.08151 (2024).

---

> > ### Comment · Reviewer_cUnp · 2024-12-02
> > **Response to authors**
> >
> > Since the rebuttal addresses all concerns I had, I decide to raise my confidence.

---

### Comment · Area_Chair_xryB · 2024-11-25

Hi Reviewers,

We are approaching the deadline for author-reviewer discussion phase. Authors didn't provide rebuttal. But feel free to ask questions.

---

### Author Response · Authors · 2024-11-26
**Summary of revision**

Dear reviewers, thank you for the detailed and constructive feedback! It would be much appreciated if we could get your feedback about whether we have addressed your concerns. We detail the following changes made in the revision:

* We updated the pseudo-code of our DyMe bank updating in Appendix A 5.2.
* We updated the analysis of memory consumption with larger video resolutions in Appendix A 8.1.
* We updated the ablation of the similarity threshold of feature fusion in Appendix A 8.2.
* We updated the ablation of the bank updating interval in Appendix A 8.3.
* We updated the ablation of the sampling strategy of DyMe bank in Appendix A 8.4.

---

### Meta-Review · Area_Chair_xryB · 2024-12-20

**Metareview:**

This paper proposed SteamV2V, a diffusion model that could do real-time streaming video-to-video translation with user prompts. Authors achieved this by maintaining a feature bank. The proposed solution is faster than previous work while maintaining temporal consistency.

7 reviewers unanimously thought this paper passed the acceptance bar.

The strengths of this paper are: 1) motivation is clear and problem is important; 2) methods are flexible and suitable for creative applications and achieves good results; 3) experimental results are solid.

weaknesses are: 1) using the feature bank to generate higher resolution could lead to a drastic increase in memory usage; 2) Some magic constants are reported without explanation; 3) The user study is presented without the necessary details; 4) The choice of SDEdit (which is 3 years old) should be explained -- why not newer methods; 5) There are still some artifacts in the result videos; 6) lags behind state-of-the-art video translation models in CLIP score and subjective evaluations; 7) The model sometimes struggles to maintain temporal consistency when handling videos with significant camera or object movement.

During the rebuttal, authors addressed reviewers' concerns. Some reviewers raised their confidence or score.

Given these, AC decided to accept the paper.

**Additional Comments On Reviewer Discussion:**

During the rebuttal, authors addressed reviewers' concerns. Some reviewers raised their confidence or score.

Given these, AC decided to accept the paper.

---

### Decision · Program_Chairs · 2025-01-22

Accept (Poster)

---

> ### Public Comment · ~Feng_Liang3 · 2025-02-05
> **Any guidance on how to resolve the ethics review**
>
> Hi,
>
> Thanks for reviewing our paper. I haven't heard from the AC or PC yet. Any guidance on how to resolve the ethics review?